# HEART: Heart Expert Assistant with ReTrieval-augmented

**Junhao Guo [1], XueFeng Shan[3*], Guoming Wang[1†], Dong Chen[2], Rongxing Lu[4], Siliang Tang[2]**

[1] Digitamedia Computing  Design lab, School of Software Technology, Zhejiang University, China
[2] Digitamedia Computing  Design lab, College of Computer Science and Technology, Zhejiang University, China
[3] Xinjiang Medical University, China
[4] University of New Brunswick, Canada

22351300@zju.edu.cn, xuefengshan@xjmu.edu.cn,{NB21013,chendongcs}@zju.edu.cn, RLU1@unb.ca, siliang@zju.edu.cn

## Abstract

As the incidence of cardiovascular diseases continues to rise, there is an increasing focus on the detection and treatment of cardiovascular diseases. However, in economically disadvantaged areas, the scarcity of medical resources make the early detection of cardiovascular diseases particularly challenging. Thus, the HEART (HEART Expert Assistant with Retrieval-augmented) model is proposed, which leverages the powerful logical reasoning capabilities of Large Language Models (LLMs) to assess whether patients have heart disease. Specifically, HEART operates on a dual-component structure, consisting of a Diagnostic Module and a Case Retrieval Module. For the Diagnostic Module, the LLM is pre-trained on a cardiac ultrasound assessment dataset to master the relevant evaluation techniques. As for the Case Retrieval Module, a text encoder transforms input cases into hidden features, which are then used to retrieve auxiliary cases. The input case and auxiliary cases are merged through a Case Fusion Layer to obtain the fused case features. Then, they are combined with prompts for inference. We have tested our model on a congenital disease dataset and achieved encouraging results. The proposed HEART model has shown tremendous potential in becoming the foundational model for predicting cardiovascular diseases.

## Introduction

Cardiovascular diseases, which encompass coronary heart disease, rheumatic heart disease, and congenital heart disease, among others, constitute a complex spectrum of disorders impacting the heart and vascular system (Obied et al. 2023). Data from the World Health Organization (WHO) indicate that cardiovascular diseases are responsible for over 17.9 million deaths annually, representing 32% of all global deaths (Institute for Health Metrics and Evaluation (IHME) 2019). This makes cardiovascular diseases a major health problem worldwide (Ghorbani et al. 2020). Fortunately, early intervention can significantly reduce mortality rates associated with these diseases.

Diagnosing and managing cardiovascular diseases are particularly challenging due to the heart's complexity and variability, requiring doctors to have extensive clinical experience (Wiegers et al. 2019). However, in some economically disadvantaged regions, the mortality rates from cardiovascular diseases are significantly higher than the average. This is mainly due to the scarcity of medical resources, particularly the lack of cardiologists, often resulting in incorrect diagnoses or unpredictability in early disease detection, all of which endanger lives (Obied et al. 2023). Nevertheless, the advent of artificial intelligence (AI) technologies offers promising solutions. AI can analyze vast arrays of cases, providing insights that aid clinicians in making more precise decisions. Echocardiography stands as one of the primary modalities for cardiac examination (Kareem and Obied 2021; Papolos et al. 2016), and current methodologies are significantly focused on the detailed analysis of heart ultrasound images (Muhtaseb and Yaqub 2022; Blaivas and Blaivas 2022; M Alaa, Philippakis, and Sontag 2022). However, these techniques generally rely on intricate preprocessing procedures and substantial computational resources. Additionally, the heart's constant motion during its beat cycle complicates the analysis of ultrasound images, as subtle morphological changes, like the radial expansion and lengthwise contraction of the left ventricle, are difficult to discern, even by experienced clinicians (Wahlang et al. 2021). And variations in cardiac structure and function also exist across different ethnicities and regions (Cohen et al. 2010; Havranek et al. 2015).

Remarkably, echocardiographic examination records, which provide detailed dynamic cardiac data documented by radiologists, have been largely overlooked in machine learning research. These records encompass critical measurements, such as the left ventricular end-systolic and end-diastolic diameters, septal thickness, and mitral regurgitation (as shown in Table 1), and are vital for clinicians to evaluate cardiac function and diagnose cardiovascular diseases. Building on this premise, we seek to leverage these cardiac examination data to enable more precise disease prediction, with the hope that these predictions can be employed in the early screening of heart diseases. By leveraging the powerful logical reasoning capabilities of LLMs (Achiam et al. 2023; Touvron et al. 2023a; Chowdhery et al. 2023), we can conduct further studies based on patients' echocardiographic examination records.

In this study, we introduce a novel model named HEART

---

*These authors contributed equally.

†Corresponding author

(Heart Expert Assistant with ReTrieval-augmented), aimed at the assessment of cardiac diseases. This model adopts a dual-phase training methodology to optimize the diagnostic process of cardiac conditions. The initial phase is aimed at giving the LLM a deep understanding of echocardiography knowledge, thereby fostering the model's foundational capability in the evaluation of echocardiographic examination records. Following this, the second phase involves task-specific fine-tuning, incorporating a retrieval-augmented strategy, and introducing a Case Fusion Layer. Case Fusion Layer efficiently integrates multiple related cases, enhancing the model's ability to recognize and analyze echocardiographic examination records. To ensure the practical application value of our model, we constructed and validated a real dataset consisting of echocardiographic examination records, specifically targeting the detection of congenital heart diseases. We believe that the proposed model will not only play a pivotal role in the field of congenital heart disease detection but also serve as a foundational model in the domain of cardiovascular disease diagnostics, applicable to a wide array of cardiac-related diagnostic and therapeutic tasks.

## Method

In this section, we will provide a detailed introduction to the proposed method, HEART, which leverages knowledge acquired from guidelines for the evaluation of echocardiography to facilitate the diagnosis of heart diseases. We will begin by describing how we created our dataset for cardiac ultrasound detection, followed by an introduction to the architecture of the proposed model.

### Data Preparing

**Pre-training**  The purpose of pre-training is to enable the foundational model to learn relevant prior knowledge from a vast corpus of information. This process involves initially pre-training the model on a large volume of unlabeled data, and then applying it to tasks within specific domains to achieve improved outcomes. In this context, we aim for the proposed model to acquire some knowledge related to echocardiography before engaging in specific tasks, thereby refining its diagnostic precision using the provided data. To this end, we have curated an echocardiography-focused pre-training dataset, which extracts textual data from relevant echocardiographic assessment reports (Silvestry et al. 2015; Lai et al. 2006; Mitchell et al. 2019; Liu and Xiong 2022). Considering the model's limitation in processing image data, we employ regularization to detect and omit sentences that analyze images. Specifically, we detect the word "Figure" in sentences and remove the corresponding sentences.

**Heart disease dataset**  To further enhance the model's capability in cardiovascular disease diagnosis, we collected an additional dataset of congenital heart diseases provided by medical institutions. In this dataset, all the cardiac conditions of the patients were categorized into three distinct classifications: (1) Atrial Septal Defect, (2) Ventricular Septal Defect, and (3) Patent Ductus Ovale. Each case presented with at least one type of congenital cardiac anomaly. The

| | |
|---|---|
| Left Ventricular End-Diastolic Diameter | Aortic Sinus |
| Left Ventricular End-Systolic Diameter | Aortic Annulus |
| Left Ventricular Posterior Wall Thickness | Pulmonary Artery |
| Interventricular Septal Thickness | Right Atrial Diameter |
| Right Ventricular Outflow Tract | Right Ventricular Diameter |
| MV_E (Mitral Valve E wave) | MR (Mitral Regurgitation) |
| MV_A (Mitral Valve A wave) | TR (Tricuspid Regurgitation) |
| MVE_E (Early Diastolic Velocity) | AR (Aortic Regurgitation) |
| FS (Fractional Shortening of Left Ventricle) | PR (Pulmonary Regurgitation) |
| EF (Ejection Fraction of Left Ventricle) | CO (Cardiac Output) |
| Left Atrial Diameter | SV (Stroke Volume) |
| MVE/A Ratio | E/A Ratio |
| Ventricular Wall Motion Score | |

Table 1: All data in the echocardiographic examination records.

challenge of this task lies in accurately predicting all cardiac defects of a patient, whether singular or multiple. We have collected a total of 1006 entries from the real echocardiographic examination records of the hospital. These entries have undergone rigorous processing, including the treatment of missing values, standardization of units, and the elimination of outliers.

### Model Architecture

As shown in Figure 1, HEART comprises two primary components: the Diagnostic Module and the Case Retrieval Module. Initially, the input case is processed by the Case Retrieval Module, which identifies the $K$ most pertinent auxiliary cases. Subsequently, these cases, along with the input case, constitute a case pair denoted as $(T, R)$. This case pair is then fed into the Diagnostic Module, where reasoning is performed on the pair $(T, R)$. In this section, we will elaborate in detail on the contents of these two modules.

**Case Retrieval Module**  We expect that when HEART is applied to cardiovascular disease diagnosis tasks, it relies not solely on reasoning from learned cases but utilizes retrieval augmentation techniques to fetch similar examples from a historical case database, thereby assisting the model in making more accurate inferences. The Case Retrieval Module consists of a case database and a text-encoder.

In the construction of the case database, our objective is for the cases in the training dataset not only to teach the model how to predict diseases but also to guide the model during the inference stage. The advancement in vector databases has enabled us to implement the detection of analogous cases effectively. For this purpose, we encode the training dataset and store it in the vector database. Specifically, for each training case $T$, HEART first tokenizes this text into subwords $\{x_{cls}, x_1, ..., x_n\}$ and transforms it into hidden representations $\{w_{cls}, w_1, ..., w_n\}$ using a text-encoder, where $n$ denotes the length of the input tokens. And then the cases in the training dataset are stored in the Faiss vector database (Douze et al. 2024) using $w_{cls}$ as the storage vector.

During the training and inference phases, each case is first transformed into hidden representations $\{w_{cls}, w_1, ..., w_n\}$ by the text-encoder. Then, the $w_{cls}$ representation is then compared with the data stored in the Faiss vector database using cosine similarity, and the top $2K$ highest scoring cases

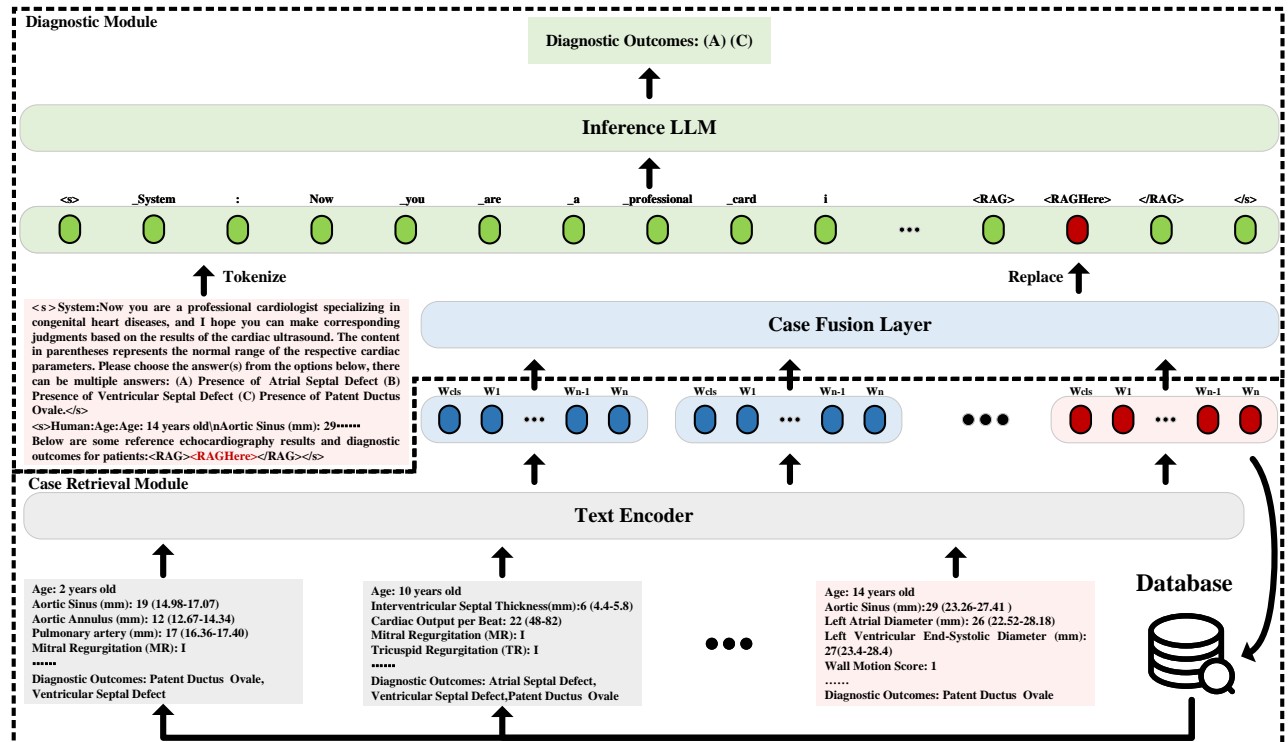

Figure 1: This picture of the proposed HEART model in detail. Upon receiving an inference case, the text-encoder encodes it into hidden features and retrieves related cases from the database, which are also encoded into hidden features. These encoded features are then input alongside the original case into the Case Fusion Layer. The resulting fused features replace the $<RAGHere>$ token in the prompt, and the combined embedding is input into the Inference LLM for answer prediction.

are retrieved. Following this, a reranker model reorders these cases, and the top $K$ cases with the highest scores are selected as auxiliary cases for assistance. At this point, the input case and the retrieved $K$ cases are combined into a case pair $(T, R)$, where $T$ denotes the input case, and $R$ represents the retrieved cases set, with $r_i \in R, i \in [0, K)$, signifying the retrieved cases.

**Diagnostic Module**   Upon processing through the Case Retrieval Module, a case pair $(T, R)$ is obtained. The Diagnostic Module is composed of two principal components: the Case Fusion Layer and the Inference LLM.

The extant models, limited by the pre-training window size, face challenges in retrieving a substantial number of cases simultaneously, which is crucial for enhancing inference accuracy. To mitigate the model's token limitations, a Case Fusion Layer has been integrated into the Inference LLM. For each retrieved case, the hidden vectors is represented as $w_i = \{w_{cls}^i, w_1^i, ..., w_n^i\}$, where $i$ denotes the $i - th$ retrieval vector, with earlier retrieval vectors having higher similarity scores. The input case's representations are denoted as $t = \{t_{cls}, t_1, ..., t_n\}$. A cross-attention mechanism is employed to amalgamate these two representation sets. The attention module in the Transformer (Vaswani et al. 2017) is defined as follows:

$$Attn(Q, K, V) = softmax\left(\frac{QK^T}{\sqrt{d_k}}\right) V$$

Using this denotation, cross-attention is defined as:

$$Q = W_Q t_{cls}, K = W_K A, V = W_K A$$

where $A = [W_{cls}^0, W_1^0, W_2^0, ..., W_n^0, W_{cls}^1, W_1^1, ..., W_n^1, W_{cls}^{K-1}, W_1^{K-1}, ..., W_n^{K-1}]$ represents the concatenation of multiple retrieved sentences into one sentence according to the order of tokens, $t_i, w_i^k \in R^{1xh}$, where $h$ denotes the representation of hidden features of the text encoder.

Then, the input prompt and auxiliary casse are amalgamated. The Inference LLM deconstructs this combination into tokens, which are then transformed into embedding vectors $P = \{p_{}, p_{System}, ..., p_{<RAGHere>}, ..., p_{}\}$. Within the input prompt, we have added three special tokens, $< RAG >< RAGHere >< /RAG >$, to denote the replace position in the input prompt. After obtaining the embedding vectors, $p_{<RAGHere>}$ is replaced with the Case Fusion vector, which is acquired from the Case Fusion Layer. Following this substitution, the embeddings are inputted into the Diagnostic Module for prediction.

During the pretraining phase, next-token prediction strategies are used to train the initial model to learn the corresponding task-specific knowledge, while in the fine-tuning phase, the loss is computed solely based on the assistant's output, refining the model's focus on the generated response.

|  | All | | Single | | | Double | | | Triple | | |
|---|---|---|---|---|---|---|---|---|---|---|---|
|  | Correct | F1-Score | Precision | Recall | F1-Score | Precision | Recall | F1-Score | Precision | Recall | F1-Score |
| *Llama2* | | | | | | | | | | | |
| Few-shot5 | 33.46% | 33.86% | 0.00% | 0.00% | 0.00% | 29.89% | 72.37% | 42.31% | 42.11% | 100.00% | 59.26% |
| Fine-tuning | 60.00% | 53.80% | 65.68% | 73.03% | 69.16% | 46.75% | 47.37% | 47.06% | 64.29% | 28.12% | 39.13% |
| w/o stanard | 45.38% | 37.59% | 64.46% | 51.32% | 57.14% | 29.01% | 50.00% | 36.71% | 25.00% | 6.25% | 10.00% |
| *RAG* | | | | | | | | | | | |
| RAG-1 | 71.15% | 78.16% | 72.57% | 53.95% | 61.89% | 70.30% | 93.42% | 80.23% | 80.00% | 100.00% | 88.89% |
| RAG-2 | 72.69% | 77.51% | 79.09% | 57.24% | 66.41% | 66.04% | 92.11% | 76.92% | 72.73% | 100.00% | 84.21% |
| RAG-3 | oot | oot | oot | oot | oot | oot | oot | oot | oot | oot | oot |
| *Proposed* | | | | | | | | | | | |
| HEART | 79.23% | 84.34% | 79.41% | 71.05% | 75.00% | 72.34% | 89.47% | 80.00% | 100.00% | 93.75% | 96.77% |

Table 2: Results for congenital heart disease detection: 'Correct' indicates the accuracy of completely correct answers. 'Single/-Double/Triple' shows the accuracy for cases with single, double, and triple choices, respectively. 'RAG-k' denotes the retrieval of k related cases per input case. 'oot' means 'out of token limit', indicating inability to the perform normal reasoning.

## Experiments

In this section, we explore the specifics of our experimental model, elucidating the configuration, implementation, and principal outcomes. Our experimental design is meticulously crafted to comprehensively evaluate the efficacy of the HEART.

## Implementation Details

**Pre-training Setting** We employ Llama2-Chinese-13b-Chat as our foundational model, which has been further fine-tuned on Chinese data atop llama2-13B (Touvron et al. 2023b). The model is trained for 10 epochs with a batch size of 64. For optimization, we utilize AdamW (Loshchilov and Hutter 2017) with a learning rate of 1e-4. We set the warm-up steps to 1000 and the token block size to 2048.

**Fine-tuning Setting** Building upon previous works, we utilize the model obtained from the Pre-training phase as the Inference Model. To circumvent feature fusion in the Case Retrieval Module, the text-encoder in the Case Retrieval Module is also configured to this model. We employ bge-reranker-large (Xiao et al. 2023) as the reranker model to reorder the input data. For each instance, we retrieve $K = 5$ cases from the database. During training, we randomly mask $m \in [0, 4]$ cases. This approach is adopted to ensure that the model pays attention to cases positioned differently. The model is trained for 15 epochs with a batch size of 4. For optimization, we utilize AdamW with a learning rate of 1e-5. We utilize LoRA (Hu et al. 2021) to accelerate our training process, with lora_alph=32, lora_r=8.

## Results

As shown in Table 2, the effectiveness of our proposed HEART model is substantiated through empirical evaluation. In the realm of congenital heart disease prediction using the Llama2 model, it is observed that while the few-shot5 inference approach is capable of generating predictions, the accuracy of model inference remains markedly low. However, a notable improvement in prediction accuracy, amounting to 26.54%, is witnessed upon fine-tuning the model. In this context, the standard values (As shown in Figure 1) of indicators in cardiac ultrasound reports play a pivotal role. In the absence of these indicators, a significant decline in the accuracy of Llama2 is discernible, particularly in tasks involving triple choices.

Subsequently, the implementation of the RAG model for case retrieval markedly enhances performance, elevating the accuracy from 60.00% to 71.15% when a single case is retrieved for assistance, and gradually increases with the growing number of retrieved cases. However, retrieving three cases poses challenges due to exceeding the token length threshold, which hampers prediction capabilities. Contrarily, the HEART model successfully overcomes token limitations, achieving an impressive accuracy rate of 79.23% with five retrieved cases.

Notably, under the few-shot5 mode, the model achieves a recall of 100% in triple choice scenarios, yet the F1-Score remains substantially low. Furthermore, the model demonstrates complete incapacity for rational inference in single-choice cases, with an F1-score of 0%. We think this is because the LLMs are too conservative in their disease reasoning task. The introduction of fine-tuning ameliorates this issue, and the HEART model transcends these constraints, exhibiting encouraging results in predicting single, double, and triple diseases.

## Conclusion

In this study, we introduce an innovative foundational medical model specifically designed for cardiovascular disease screening, named HEART. This model is focused on processing and analyzing basic cardiac report data, which are readily obtainable and storable during cardiac examinations, to assist inexperienced clinicians in making more accurate diagnoses. By employing retrieval-augmented techniques, the HEART model significantly improves the accuracy of cardiovascular disease prediction. In the diagnostic task for congenital heart disease, the HEART model achieved an accuracy rate of 79.23%, markedly surpassing the benchmark model's accuracy of 45.38%. We believe that the HEART model can become a cornerstone in the field of cardiovascu-

lar disease screening and that its application can be extended to a broader range of diagnostic tasks related to cardiovascular diseases.

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
