# OpenReview forum: "HEART: Heart Expert Assistant with ReTrieval-augmented"
_AAAI.org/2024/Spring_Symposium_Series/Clinical_FMs — AAAI 2024 SSS on Clinical FMs_

### Official Review · Reviewer_LEQu · 2024-02-17
**Using ECG examination records to predict heart defects**

**Rating:** 4
**Confidence:** 3

**Review:**

The authors propose to use ECG examination records (text) to classify heart defects. This is in contrast to the more "classical" approach of using directly ECG data.
To my understanding, at inference time this method would rely on a doctor first describing the ECG to produce the ECG examination record which would then be used as input to the model. I wonder if this is something that limits its applicability. In the introduction the authors note that there is a lack of cardiologists, but this method would not alleviate that issue.
The authors first apply a classic few-shot strategy using a llama2 variant that was pre-trained on a curated version of a public dataset of ECG notes. They note a low performance with this strategy.
Then they finetune the model on their dataset (1006 cases) and observe a marked improvement. Notably, there are no descriptions of whether they split the data in training and validation or of any cross-validation strategies. Additionally, pre-training and finetuning are done for very few epochs 10 and 15 respectively.
Then the authors use RAG to include context from similar cases to aid in the prediction. They can only include 1 or 2 retrieved samples before running out of tokens with a standard RAG strategy. The RAG strategies improve the performance. But I have several issues here. i) The knowledgebase for RAG is the same training cohort. This seems a major issue. ii) If the retrieved context includes a diagnosis, the model may just use the retrieved diagnosis. It may be beneficial to investigate this potential issue.
To include more context, they propose a context fusion model based on cross-attention. Then the entire context coming from the RAG portion boils down to a single vector if I understood correctly. This seems to improve performance further.
In general I would have liked to see baselines of performance using ECG data to compare if there could be benefits of the LLM strategies based on ECG reports.
I was confused about the "standard values" I could not find a description of them. They are mentioned only once in the text but they appear in the results table.
It would have been good to see the number of cases in each type of case: single, double, triple.

---

### Official Review · Reviewer_Ek3d · 2024-02-18
**Good proposed approach but need further polish**

**Rating:** 5
**Confidence:** 4

**Review:**

The manuscript proposes a retrieval-augmented LLM approach for text-based cardiovascular disease detection.
Overall, the proposed approach is reasonable. However, there exists a number of unclear aspects that need to be addressed before the manuscript to be published.

**Strengthese**
- cardiovascular disease detection is one of the important risk prediction scenarios for clinical foundation models.
- the proposed retrieval augmentation is a useful technical to enhance foundation models

**Weaknesses**
- Heart disease dataset
  - since it is an author-collected dataset, it is better to mention the size of training and testing set
  - in the real clinical setting, there can exist healthy patients. However, the dataset does not contain the label for the healthy condition.
- Technical part
  - how do the authors implement the re-ranker? Now there is no explanation for that.
  - how to derive ``<RAGHere>`` from ``A``? In the introduction of the case fusion layer, the authors stop after introducing their cross-attention operator. There still exists a gap between the cross-attention and ``<RAGHere>``
  - is there a particular reason to use the same $W_K$ for both $K=W_K A$ and $V=W_K A$?
- Experiment setup
  - what is the RAG model? there is no reference for it. If it is a custom baseline, it would be better to introduce it.
  - "During training, we randomly mask $m$ cases." $m$ is first introduced here. This introduction of a new variable without prior explanation can pose challenges for understanding the method effectively.

**Questions**
- Can the authors explain why the retrieved cases are still from the training data? Assuming the foundation model is well-trained, then it does not need
- what do ``standard values`` refer to (Results section)? Also, there is a typo for ``w/o stanard`` in Tabl 2. Is the ``standard value`` similar to the concept of ``standardization of units`` mentioned in the Heart disease dataset section? I can guess it may refer to the standard value range of a vital, but it is better to explicitly introduce it.

---

### Official Review · Reviewer_6pZo · 2024-02-22
**Strong Manuscript**

**Rating:** 9
**Confidence:** 4

**Review:**

In reviewing this manuscript, it is clear the authors can intellectually articulate the focus of their study. They were able to explain the rationale of their research as well as their results.

My constructive feedback would be as follows:

1. The abstract should be more concrete in explaining the “encouraging results.” At present, there is no concrete description of any results whatsoever in the abstract.

2. In defining the attention function, the authors should define all terms in the model; at present, they do not describe the \\(d_k\\)
scaling term and the role it plays in the attention function.

3.  In the section about pretraining, the authors state, "Specifically, we detect the word “Figure” in sentences and remove the corresponding sentences." Without having looked at the curated pre-training dataset, it would be imperative to know if all sentences analyzing images had the word "Figure" or words such as "Image" or other synonyms might have been used.

4. I would encourage the authors to explicitly define the "cls" subscript used in their arrays.

5. When describing the space of retrieved sets, I believe the authors may have a typesetting issue; namely, the authors state the set as \\(R^{1xh}\\), using an italic \\(x\\) as opposed to \\(\\times\\), indicating the array size of R being 1 by h. Hence, I believe the dimensionality of R should be represented as \\(R^{1 \\times h}\\).

6. Since the authors are using LaTex markups, I encourage them to express the learning rate as \\(10^{-4}\\) as opposed to 1e-4

7. The authors should define lora_alph and lora_r in the context of LoRA and display them with the appropriate LaTex markup (if applicable).

Overall, the authors have an extremely strong paper.

---

### Official Review · Reviewer_sCWy · 2024-02-26
**A nice end-to-end pipeline showing LLMs being used to assist in clinical predictive tasks**

**Rating:** 7
**Confidence:** 4

**Review:**

Well written paper proposing a pipeline capable of predicting cardiovascular diseases. Custom models are used with a RAG layer to retrieve predictions.

The evaluation is well thought out. Overall a good contribution.